# Rutin, a Flavonoid Compound Derived from Garlic, as a Potential Immunomodulatory and Anti-Inflammatory Agent against Murine Schistosomiasis *mansoni*

**DOI:** 10.3390/nu15051206

**Published:** 2023-02-28

**Authors:** Rabab S. Hamad

**Affiliations:** 1Biological Sciences Department, College of Science, King Faisal University, Al Ahsa 31982, Saudi Arabia; rhamad@kfu.edu.sa; 2Central Laboratory, Theodor Bilharz Research Institute, Giza 12411, Egypt

**Keywords:** rutin, garlic, schistosomiasis, praziquantel, antioxidant

## Abstract

Schistosomiasis is a tropical disease caused by trematode worms. The inflammatory response of the host to schistosome eggs leads to formation of granuloma in the liver and intestine. Praziquantel (PZQ) is still an effective treatment for schistosomiasis, however resistance development may reduce its efficacy. The current study investigated the possible immunomodulatory and anti-inflammatory action of rutin, a natural flavonoid compound isolated from garlic, on liver fibrotic markers in mice infected with *S. mansoni* in comparison to PZQ. Male albino CD1 mice were infected with 100 ± 2 *S. mansoni* cercariae/mouse and treated with garlic, rutin, or PZQ. At the end of the experiment, the liver and intestines were harvested for parasitological and histological assessment and to analyze the proinflammatory cytokine. Rutin significantly affects the pathological alterations caused by *Schistosoma* in the liver. This may be partially explained by a decrease in the number of eggs trapped in the tissues of the liver and a modification in the serum levels of certain cytokines, which are implicated in the formation of *Schistosoma* granuloma. In conclusion, rutin has strong anti-schistosome properties in vivo, raising the possibility that rutin might be further investigated as a therapy for *S. mansoni*.

## 1. Introduction

Schistosomiasis is an infectious parasitic disease caused by trematode parasites of the genus *Schistosoma*. Schistosomiasis affects more than 200 million individuals worldwide. About 200,000 people per year die from the disease, which is endemic mostly in economically developing parts of Asia, South America, and sub-Saharan Africa [1,2,3]. *Schistosoma* (*S.*) *mansoni*, *S. haematobium*, and *S. japonicum* are the most common human-related schistosomiasis species [4].

Schistosomiasis had an impact on a number of variables. *S. mansoni* caused an increase in the serum lipid profile and liver functioning [5], the eggs are first deposited in the infected host’s vasculature by adult female *S. mansoni*; then, it begins to induce a granulomatous inflammatory response [6]. As a result, pathological disorders in these tissues occur. *S. mansoni* increased the levels of lipid peroxide, indicating an increase in oxidative stress [7].

Innate and adaptive immune responses are involved in the parasite immunological response to schistosomal antigens. Th1-type immune responses, which are predominant in the acute phase of schistosome infection, release cytokines such as IFN-γ, IL-12, and TNF-α that can prevent pathogen invasion by destroying it [8,9,10]. Granuloma and liver fibrosis are also somewhat inhibited by the Th1-type immune response. The host’s immune response changes to a Th2-type after 4 weeks of infection and starts to produce cytokines including IL-4, IL-10, and IL-13. The size of the egg granuloma decreases as the infection moves into the chronic phase, Treg cells become active, and the Th1/Th2 immunological balance is controlled and maintained, all of which prevent the growth of liver fibrotic lesions [11,12]. Egg granulomatous lesions can be significantly reduced by using IL-17 neutralizing antibodies since Th17 cells are directly connected with pathological liver damage [13].

Praziquantel (PZQ) is the main anthelmintic drug used to control schistosomiasis; nevertheless, numerous reports have shown that PZQ might cause an immunological response that increases resistance to re-infection [12,14]. As a result, there has recently been an upsurge in demand for innovative and potent anti-schistosomal drugs, particularly those made from natural ingredients.

Numerous studies suggested that *Allium sativum* (garlic) consumption may help prevent the development of chronic diseases [15,16]. Previous studies highlighted the anti-inflammatory and immunoregulatory properties of garlic in addition to its powerful antioxidant activity. These health advantages of garlic come from its polyphenols and organosulfur components [17]. Several studies examined the effectiveness of using garlic or its aqueous or oil extracts as a treatment for Schistosomiasis *mansoni* [18,19,20,21].

Rutin is a polyphenolic flavonoid that is abundantly found in garlic as well as in different food sources. In recent years, a number of studies investigated the diverse biological effects of rutin, such as anti-microbial, anti-carcinogenic, anti-thrombotic, cardioprotective, and neuroprotective activities along with improving the liver’s health, function, and integrity [22,23,24]. These biological advantages are mostly attributable to its anti-inflammatory and antioxidant properties [25].

To the best of our knowledge, no research has been done on how rutin, isolated and purified from garlic, affects schistosomiasis. Therefore, this study aims to compare the possible anti-inflammatory, immunomodulatory, and parasitological efficacy of the garlic ingredient rutin, to PZQ in preventing pathological alterations brought on by *S. mansoni* infections.

## 2. Materials and Methods

### 2.1. Plant Material

*Allium sativum* (garlic) bulbs (*AS*b) were collected from a private farm, Al-Ahsa, Kingdom of Saudi Arabia, in the summer of 2020. The fresh *As*b were cleaned, washed, dried in shade then in oven at approximately 50 °C till the constant weight, then ground to a fine powder, sieved (80-mesh sieves), and prepared for use in the experiments.

### 2.2. Preparation of ASb Extract

ASb extraction was carried out as per our prior investigation [21]. Briefly, five kilograms of the *AS* bulbs were extracted with 70% aqueous ethanol extract using the soxhlet apparatus at 80 °C until the solution became clear (~72 h), and then the aqueous ethanol extract was filtrated and evaporated under reduced pressure till very small volume. Finally, a sticky dark brown extract yielded 132 g. 

### 2.3. Determination of Total Phenolic and Total Flavonoid Content

The total phenolic contents of the *AS*b extract were spectrophotometrically estimated according to the Folin–Ciocalteu (FC) colorimetric reagent and the total flavonoids were determined according to Karawya and Aboutabi [26].

### 2.4. Investigations of Flavonoids

#### 2.4.1. Separation and Identification of Rutin Compound

The concentrated *AS*b ethanolic extract was chromatographed separately on paper chromatography (PC) Whatman No. 1 using the solvent system n-Butanol: Acetic acid: water (4:1:5 *v*/*v*/*v*) (BAW) (S1) for the first way and solvent system Acetic acid: water (15:85 *v*/*v* AcOH-15%) (S2) for the second way, air-dried, and examined under ultraviolet (UV) light, then exposed to ammonia and re-examined under UV light to observe the possible changes that may eventually appear in color or fluorescence. The previous steps of the examination were applied for column chromatography, where column chromatography remains the single most useful technique for the isolation of large quantities of flavonoids from crude plant extract. The concentrated *AS*b ethanolic extract (dry method) was applied on the top of a polyamide column. Elution was started with 80% ethanol. The received fractions were evaporated and subjected to PC where similar fractions were collected together [27]. Each fraction was separated when subjected to preparative paper chromatography (PPC) Whatman No 3 PC using solvent system S1 and S2, where the developed chromatograms were air-dried and examined under UV light. The major separated flavonoid compounds were purified on sephadex LH-20 column chromatography. The pure flavonoid compound was identified by chemical method and confirmed by physical method.

#### 2.4.2. Chemical Analysis

Controlled (mild) acid hydrolysis, complete (normal) acid hydrolysis according to Harborne et al. [28].

#### 2.4.3. Physical Analysis

Ultraviolet spectrophotometer analysis (UV):

Chromatographically pure materials were dissolved in pure methanol and subjected to ultraviolet spectrophotometric investigation using shimadzu UV-240 spectrophotometric analysis and addition of the different reagents [29].

^1^ H- NMR: 

The NMR measurements were carried out on JEOL EX-270 NMR spectrometer apparatus 270 MHz for ^1^ H- NMR.

### 2.5. Parasites and Animals

Fifty male albino CD-1 mice, 6–8 weeks old (weight ~25 g), were maintained and bred at animal unit, Theodore Bilharz Research Institute (TBRI). Mice were maintained under a controlled condition with free access to water and standard commercial pellet diet. 

*S. mansoni* cercariae were purchased from TBRI and used directly after shedding from *Biomphalaria alexandrina* snails. CD1 mice were infected via the tail by 100 ± 2 *S. mansoni* cercariae/mouse [30].

The animal experiments were carried out according to the animal ethics guidelines after approval of the Scientific Research Ethics Committee of King Faisal University under approval number KFU-REC-2022-JAN-EA000363.

### 2.6. Study of Acute Oral Toxicity of Rutin in Mice

To investigate the acute toxicity of rutin, a study was performed to choose the suitable doses for mice. Twenty male albino CD-1 mice, 6–8 weeks (weight ~25 g), were divided into 4 groups (5 mice each) and kept in clean cages for 6 days to acclimatize to the laboratory conditions. Mice were fasted for 4 hours, then rutin was administered orally at the dose of 20, 30, 40 mg/kg body weight for 14 days, while the control group received distilled water. After administration, mice were kept fasted for another 2 h and closely observed after 24, 48, and 72 h for the development of any signs of acute toxicity. Parameters for observations include alterations in behavior, regular activities, a rough coat, mobility, diarrhea, and death. Body weight measurements were taken on days 0 (before injection), 7, and 14. At the end of the experiment (day 14), the mice were anesthetized and blood was collected by cardiac puncture technique for the analysis of hematological parameters. A daily dose of 40 mg/kg/day was selected for the oral treatment in this study.

### 2.7. Experimental Design

Mice were allowed to adapt to the laboratory environment for one week before the experiment. Mice were divided into 5 groups (ten mice each) according to the experimental design shown in Table 1.

All mice were weighed then anesthetized using diethyl ether at 8 weeks postinfection and subjected to the following parameters. 

### 2.8. Parasitological Analysis

Mature worms were recovered from the hepatic portal system using the liver perfusion technique seven weeks after infection with 100 viable *S. mansoni* cercariae [32]. The recovered worm pairs were subsequently exanimated and counted under a dissecting microscope. Percent (%) change in parasite load was evaluated by the following formula:

% change = [mean number in untreated infected controls–mean number in treated infected mice/mean number in untreated infected controls] × 100.

Total egg load in both liver and feces of each mouse were evaluated by using the Kato–Katz technique [33]. Egg burdens were quantified as eggs per gram of liver or feces according to the previous formula.

### 2.9. Histopathological and Morphometric Analysis

Liver samples were weighed, immediately fixed in 10% formalin for 48 h, then liver samples were dehydrated in increasing series of alcohol, paraffin embedded and sectioned into 4 mm thick slices that were stained with hematoxylin and eosin (H&E; Merck Millipore, Guyan court, France) to prepare the histology slides [34]. The histopathological examination was implemented by optical light microscopy through scoring of inflammatory infiltration, fiber accumulation, tissue damage, and necrosis, according to the method of Shackelford et al., [35], in which tissue lesions were classified as no lesion (0), mild (1), moderate (2), or severe lesion (3).

Assessment of granuloma numbers and diameter was performed for individual mice by photographing 6 random fields from the previously prepared histology slides using Olympus photomicroscope (Olympus, Center Valley, PA, USA). Only granulomas with eggs in the center were assessed. Assessments were accomplished via the Olympus DP Controller software.

### 2.10. Serum Sample Preparation 

Blood samples were collected from different mice groups by heart acupuncture and sera were separated and centrifuged for 15 min at 2000 rpm then stored in aliquots at −80 °C until use.

### 2.11. Biochemical Analysis

#### 2.11.1. Assessment of Liver Biomarkers

Activities of serum alanine transferase (ALT), aspartate transferase (AST), and alkaline phosphatase (ALP) enzymes were measured using commercial kits (Roche, Mannheim, Germany). 

#### 2.11.2. Assessment of Antioxidant Markers

Liver tissues were perfused and homogenized and used to determine the antioxidant enzyme activities, and catalase (CAT) and superoxide dismutase (SOD) activity that were measured following the manufacturer’s instructions (Biodiagnostics, Co., Worcester, UK).

### 2.12. Cytokine Analysis

The serum levels of type 1 cytokine (tumor necrosis factor-alpha (TNF-α)), type 2 cytokine (interleukin (IL-)4), and a T helper 17 cytokine (IL-17) were evaluated using a standard sandwich ELISA kit (MyBioSource, Inc., San Diego, CA, USA). (R&D Systems Inc. and, Minneapolis, MN, USA).

### 2.13. Flow Cytometric (FCM) Analysis

For apoptosis assays, the mouse spleen was isolated and forced through a 100 μm cell strainer (BD Biosciences, San Jose, CA, USA) to obtain a single-cell suspension. Red blood cells (RBCs) were lysed by RBCs lysis buffer (Sigma, St. Louis, MO, USA) and remaining spleen cells were washed twice by a pyrogenic saline. 

Measurement of the apoptotic cell death was performed using FCM analysis. Spleen cells were washed with PBS and resuspended in ice-cold binding buffer and stained with Annexin V-FITC (5 μL) and Propidium Iodide (PI, 10 μL) in dark at 23 °C for 15 min. Thereafter, the cells exhibiting apoptosis were analyzed using a flow cytometer (Becton–Dickinson, Franklin Lakes, NJ, USA). The experiments were performed in triplicate. 

### 2.14. Statistical Analysis

Data were first tested for normality using the Shapiro–Wilk test that was non-significant, indicating normally distributed data. One-way ANOVA was used, followed by a Tukey test to determine the statistical significance by using SPSS version 20 (SPSS Inc., Chicago, IL, USA) and GraphPad Prism 8 software (GraphPad Software Inc., San Diego, CA, USA). Descriptive analyses and measures of central tendency were performed to describe the sample characteristics. Data are expressed as mean ± standard error of the mean (SEM). *p <* 0.05 was considered as significant for all statistical analyses.

## 3. Results

### 3.1. Phytochemical Analysis

ASb has total phenolic and flavonoid contents of 30.54 ± 2.12 mg/100 g and 38.86 ±1.38 mg/g dry weight, respectively.

### 3.2. Isolation and Identification of Rutin Compound

The main isolated component was visible under UV light as a purple-colored compound that turned yellow when exposed to ammonia, with R_f_ 0.61 and 0.53 in BAW and AcOH 15%, respectively, which indicated that the compound may be of a flavonoid glycoside nature [36]. Complete acid hydrolysis resulted in the production of two sugar residues known as glucose and rhamnose, as well as the aglycone quercetin (comparative R_f_—values with authentic markers). UV spectral data in Table 2 and Figure 1A showed that the compound is a flavonol with 3-OH substitution. The remaining UV spectral data were found to be similar to that of a quercetin-type compound. ^1^H-NMR data were illustrated in Table 2 and Figure 1B by comparing with those published before [29], and the compound was identified as rutin (Figure 1C) (quercetin-3-O-α L-rhamnoside (1–6) β D-glucoside).

### 3.3. Acute Oral Toxicity of Isolated Rutin in Mice

In the present study, rutin found no interactive or toxic effect on mice, there was a regular increase in body weight during the test with no significant different between all treatment groups and controls (Appendix A); in addition, rutin did not show any mortality or side effects. As shown in Appendix A, rutin therapy for 14 days at any dose showed no impact on hematological parameters. A daily dose of 40 mg/kg/day was selected for the oral treatment in this study, supposing that the highest dose would be the most successful at eliminating helminths.

### 3.4. Rutin Reduces Adult Worm S. mansoni Burden and Egg Production in Liver Tissue and Feces

Total worm burden and egg count in the liver and feces of infected and treated mice are presented in Table 3. Treatment of *S. mansoni*-infected mice with *AS*b ethanolic extract, rutin, and PZQ showed a significant reduction in the number of *S. mansoni* worms recovered from infected mice, and in the liver and feces egg count in comparison with the infected untreated group.

### 3.5. Rutin Reduces the Pathological Damage in S. mansoni-Infected CD1 Mice

We investigated the body weights and liver weights in addition to the granuloma diameter of untreated mice, infected mice, AS-, rutin-, and PZQ-treated mice, and these were analyzed to investigate tissue responses.

The body weights of mice in the rutin-treated group were considerably higher than those in the *S. mansoni*-infected mice group that was not treated (29.8 ± 0.5, *p* < 0.001 and 23.04 ± 0.62, *p* < 0.001, respectively), but there was no significant difference when compared to the normal control group (Figure 2A). Statistically significant variance in mice liver weights between groups was detected; Tukey post hoc test revealed a significant increase in liver weights in the infected group (9.2 ± 0.39, *p* < 0.001) compared to the normal control (6.45 ± 1.1). There was no significant difference between the liver weights in the AS-, rutin-, and PZQ-treated groups when compared to the normal controls. Additionally, the rutin-treated group showed no significant difference from the PZQ-treated group (Figure 2B).

Granulomas were alerted and their number decreased significantly in both rutin-treated and PZQ-treated mice (8.2 ± 1.7, *p* < 0.001 and 7.0 ± 1.7, *p* < 0.001, respectively) compared to the infected group (Figure 2C). Significant changes in granuloma diameter were observed in both rutin-treated and PZQ-treated mice (239,327.4, *p* < 0.001 and 206,027.4, *p* < 0.001, respectively) compared to the infected group (Figure 2D).

Histopathological analyses of liver tissues stained with hematoxylin and eosin revealed that the normal control group showed normal liver architecture, with central veins and portal tracts. In *S. mansoni*-infected mice, however, eggs were found trapped in the center of the liver and were surrounded by a significant increase in inflammatory infiltrates, forming fibrous granuloma. There was a noticeable improvement in granuloma attenuation with rutin treatment, which was comparable to PZQ treatment (Figure 2E).

The severity of fibrosis was also assessed using the fibrosis scoring system, which revealed that rutin, like PZQ, had similar beneficial effects on *S. mansoni*-induced fibrosis (Table 4).

### 3.6. Rutin Reduces Changes in Serum Liver Biomarkers and Antioxidant Markers in Mice

To assess the extent of hepatocyte injury, serum transaminases ALT, AST, and ALP were measured (Figure 3A–C). In comparison to normal mice, infected untreated mice showed a significant increase in the activity of both ALT and AST enzymes. Mice treated with rutin and PZQ had considerably lower ALT and AST values than *S. mansoni*-infected control mice. ALP infection resulted in a significant decrease compared to normal mice. All infected treated groups showed a significant decrease in ALP levels when compared to untreated infected mice.

The antioxidant enzymes CAT and SOD were examined in order to evaluate how schistosomiasis causes oxidative damage in hepatic tissue (Figure 3D,E). The activity of both enzymes was significantly reduced by *S. mansoni* infection when compared to those of the normal control. Rutin and PZQ treatment both significantly reduced this inhibition.

### 3.7. Rutin Induced Changes in Serum Cytokines in S. mansoni-Infected Mice

We examined the expression of Th1/Th2/Th17 cytokines, which are the main cytokines responsible for granulomatous inflammation, in order to investigate the immunomodulatory effect of rutin. ELISA showed that *S. mansoni* infection induced an apparent increase in the plasma levels of TNF-α, IL-17, and IL-4, which were dramatically reduced in the group that received rutin (Figure 4).

### 3.8. Rutin Induced Apoptosis in the Spleen

Infected mice treated with rutin also showed decreased splenic weight (Figure 5A), suggesting that rutin may have the ability to prevent splenic apoptosis. The existence of apoptosis during *S. mansoni* infection was determined by flow cytometry analysis, and it decreases following rutin treatment (Figure 5B–G).

These findings demonstrated that rutin, like PZQ, has the capacity to reverse apoptosis in the spleen of mice infected with *S. mansoni*.

## 4. Discussion

Schistosomiasis remains a serious public health issue. Praziquantel is still the drug of choice for treating schistosomiasis at this time. It successfully kills the worm, but it is unable to repair liver damage or stop reinfection. Additionally, both in vitro and in vivo investigations have revealed that praziquantel may be resistant [12,14,37]. Several investigations have shown that a variety of natural products are active against *S. mansoni*, although their exact mechanisms of action are still unknown [38,39]. Therefore, this study aims to compare the possible anti-inflammatory, immunomodulatory, and parasitological efficacy of rutin, the main *AS* ingredient, to PZQ in preventing pathological alterations brought on by *S. mansoni* infections. Our research offers the first direct experimental proof that rutin has strong anti-schistosome properties in vivo, raising the possibility that rutin might be further investigated as a therapy for *S. mansoni*.

In this investigation we isolated the bioactive component that gives Allium sativum its anti-schistosomal properties. By using structural elucidation research, the compound was identified to be rutin, which was characterized and confirmed by ultraviolet and 1H-NMR spectral analysis [40,41,42]. The isolated compound belongs to the flavonoid family. We demonstrated that rutin has no toxic effects on mice even after 14 days of treatment at any of the doses used in the present study.

Preventing the formation of eggs in the vertebrate host is one of the crucial challenges in the treatment of schistosomiasis [12]. In the current study, we observed that, similarly to PZQ therapy, rutin induced a significant and dramatic decrease in hepatic and fecal eggs burden. These findings demonstrate that rutin treatment can significantly reduce the fecundity of surviving worms.

Numerous studies have shown that mice exposed to experimental schistosomiasis *mansoni* experienced body loss or low body weight gain along with hepatosplenomegaly and a high intestinal index [12,21]. Our findings indicated that, after 56 days of infection, *S. mansoni*-infected mice showed considerable body weight reduction as well as liver and spleen enlargement. Liver enlargement could be due to a deposition of *S. mansoni* eggs. The increase in spleen weight, however, might be due to a high level of activation during the immunological response to the parasite. In this work, rutin therapy prevented weight loss and hepatosplenomegaly in *S. mansoni*-infected mice. Reduction in liver and spleen diameter may be due to the decreasing number of eggs trapped in organs following rutin’s schistosomicidal effect.

Schistosome eggs that deposit in the liver, intestines, and other organs after infection with *S. mansoni* cause severe tissue damage, including granulomatous inflammation and tissue fibrosis [3,6]. As a result, lowering the number of eggs in the tissues can considerably lessen schistosomiasis symptoms [10,38,39]. Based on histological examination of the liver of mice treated with rutin, rutin exhibited efficiency in reducing the adult worm burden and *S. mansoni* egg production in liver tissue and feces compared to the Praziquantel that was used as a positive control. In fact, when compared to the infected mice, the rutin treatment decreased all of the hepatic pathological indicators that were assessed, including the inflammatory infiltration, tissue damage, necrosis, presence of granulomas, and vascular congestion. Additionally, the granuloma area and overall number of granulomas in the liver were also reduced. Treatment with PZQ has also been shown to reduce the size of hepatic granulomas in mice infected with *S. mansoni* [12]. However, we have demonstrated that treatment with rutin had a better immunomodulatory effect, reducing IL-4 levels and subsequently the granuloma formation process, which is in line with the observations by Ahmed et al. [24], according to which improvements in liver function biomarkers following rutin treatment were connected to improved liver histological architecture and integrity.

Schistosome eggs cause liver fibrosis, a frequent medical condition that can lead to irreversible cirrhosis and render the liver incapable of performing its metabolic function [7,38]. When schistosome eggs are deposited in the liver of *S. mansoni*-infected mice, this causes a granulomatous reaction that causes the wounded hepatic cells to leak serum levels of ALT, AST, and ALP [43,44]. The increase in these enzymes in serum is most likely brought on by hepatocyte death brought on by the release of toxins from parasite eggs [12,43,44]. The results of the current investigation demonstrated that giving rutin to infected mice dramatically lowers their serum levels of ALT, AST, and ALP. Our results are similar to previous reports indicating that rutin can help in enhancing antioxidant defenses against inflammatory assaults by reducing the release of inflammatory cytokines [24,45].

Reactive oxygen species (ROS) are produced early as a result of *S. mansoni* infection, which causes oxidative stress in the liver [12,44]. Due to the depletion of several endogenous antioxidants in the current study, *S. mansoni* reduced the antioxidant defense. SOD and CAT activity, two important endogenous antioxidants, are markedly reduced in infected, untreated mice, which is an indication of liver cell damage. The increased oxidative stress and cytotoxicity of H_2_O_2_, which is created when glutathione reductase activity is inhibited, may be related to this deficiency [45]. In agreement with previous reports [24,46], we found that rutin administration appears to counteract oxidative stress and maintain the antioxidant capacity of the liver cells since it raises SOD and CAT activity levels. The hydroxyl groups at positions 5 and 7 of the A ring are the major functional groups in rutin responsible for its antioxidant activity.

As apoptosis and inflammation are frequently linked to fibrosis, it is important to note that oxidative stress and the excessive release of ROS trigger apoptosis via an intrinsic route. Therefore, using rutin to reduce oxidative stress and strengthen the antioxidant defense system may be essential for preventing apoptosis [24]. In line with this, our findings demonstrated that rutin has the capacity to reduce inflammation and apoptosis in the spleen of mice infected with *S. mansoni*.

It has been established that the immune system is crucial in *Schistosoma* infection. Liver fibrosis in schistosomiasis is caused by an increased Th2 response and a diminished Th1 response. The current findings demonstrated a decrease in cytokines produced by Th1/Th2 faculties, primarily in the rutin and PZQ groups. Increased numbers of Th17 cell populations are associated with more severe pathology in schistosomiasis and have been connected to severe hepatic inflammation [13]. These results supported our observations about IL-17 production in mice infected with *S. mansoni*. Similarly, a number of studies have shown that a Th17 response was responsible for the progression of granulomatous lesions in *S. mansoni* infection [11,12]. Rutin’s ability to modulate the immune system may be responsible for its impact on cytokine imbalances. Additionally, by directly influencing the immunopathological response and lowering the development of hepatic granulomas, the modulation of cytokine release is linked to the lymphoproliferative effects.

This study has some limitations. First, the quantity of rutin in the garlic extract was not assessed, so we were unable to compare the rutin and garlic groups with better accuracy. Second, the expression of relevant genes and proteins in the liver and spleen tissues was not examined; therefore, we recommend further research on the molecular level.

## 5. Conclusions

Our research shows that rutin therapy at a dose of 40 mg/kg body weight displayed anti-schistosomal action. Lowering the parasitological burden and hepatosplenomegaly significantly protected mice against *S. mansoni*-induced disease. It also controls the Th1, Th2, and Th17 immunological responses caused by *S. mansoni* infection in mice, as well as hepatic severe granulomatous reaction and fibrosis and hepatic and splenic oxidative stress. Therefore, in order to understand the anti-schistosomal activity, we recommend more research into the mechanism of action of rutin.

## Figures and Tables

**Figure 1 nutrients-15-01206-f001:**
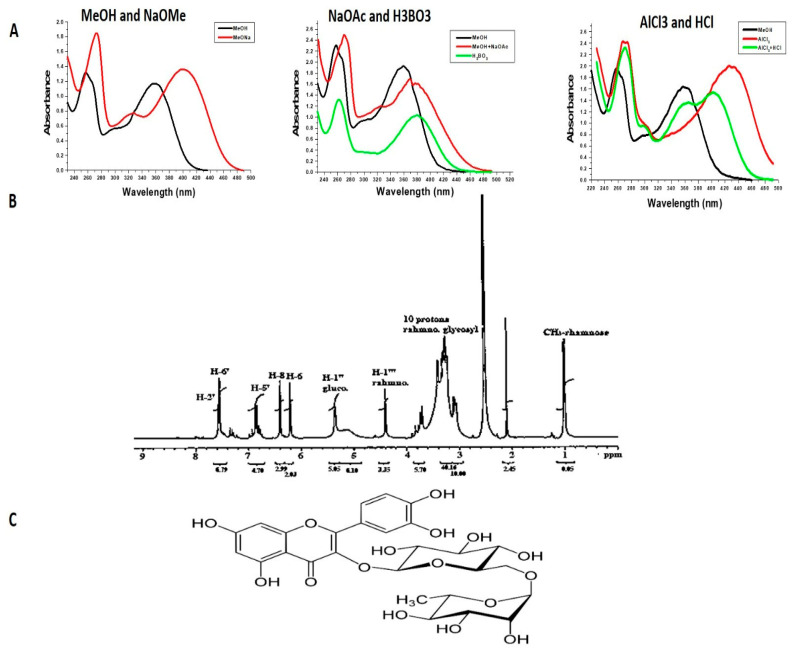
Identification of the bioactive compound. The compound rutin was identified according to the (**A**) UV spectrum in MeOH and NaOMe, NaOAc and H_3_BO_3_, AlCl_3_ and HCl; (**B**) 1H-NMR spectrum of rutin. (**C**) Chemical structure of rutin.

**Figure 2 nutrients-15-01206-f002:**
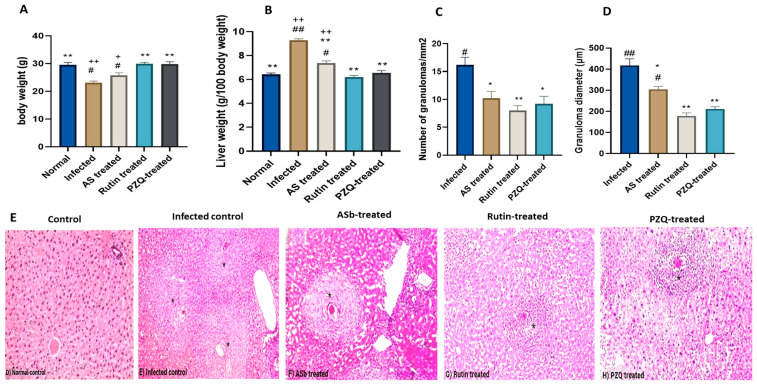
Effect of treatment with rutin naturally isolated from A. sativum on the pathological damage caused by infection with 100 ± 2 *S. mansoni* cercariae, 8 weeks postinfection. Effect of different treatments on (**A**) body weight, (**B**) liver weight (relative to body weight), (**C**) number of granulomas, (**D**) average granuloma diameter in the liver tissues of *S. mansoni*-infected CD1 mice, (**E**) histological liver sections stained with H&E (Scale bars: 100 µm). Black star shows the granuloma area. Data expressed as mean ± SE (*n* = 5). *p*-value represented the relationship between *AS*b- and rutin-treated groups and normal control, and *S. mansoni*-infected controls and drug controls. * *p* < 0.05, ** *p*-value < 0.001, compared with infected control group; + *p* < 0.05, ++ *p*-value < 0.001 in comparison to normal control group; # *p*< 0.05, ## *p*-value < 0.001 in comparison to PZQ control group.

**Figure 3 nutrients-15-01206-f003:**
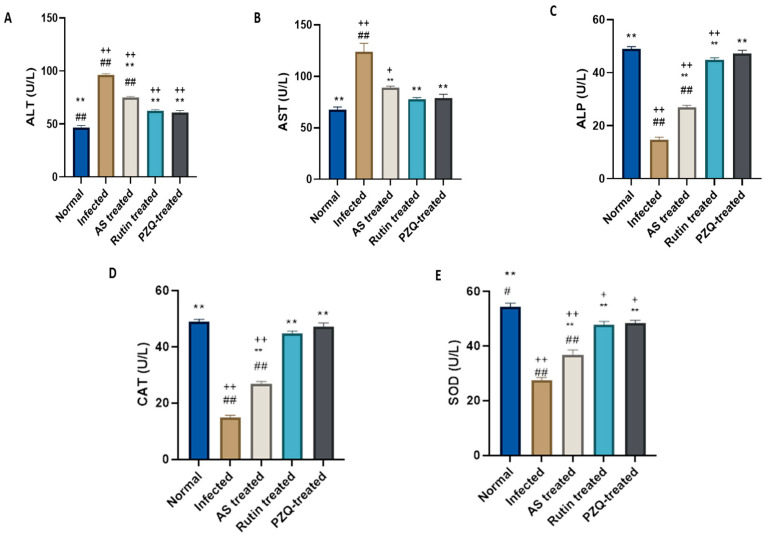
Effect of treatment with rutin naturally isolated from *A. sativum* on the liver biomarkers. Effect of different treatments on (**A**) ALT; (**B**) AST; (**C**) ALP; (**D**) CAT activity; (**E**) SOD activity. Data expressed as mean ± SE (*n* = 10). *p*-value represented the relationship between *AS*b- and rutin-treated groups and normal control, *S. mansoni*-infected controls and drug controls. ** *p*-value < 0.001, compared with infected control group; + *p* < 0.05, ++ *p*-value < 0.001 in comparison to normal control group; # *p*< 0.05, ## *p*-value < 0.001 in comparison to PZQ control group.

**Figure 4 nutrients-15-01206-f004:**
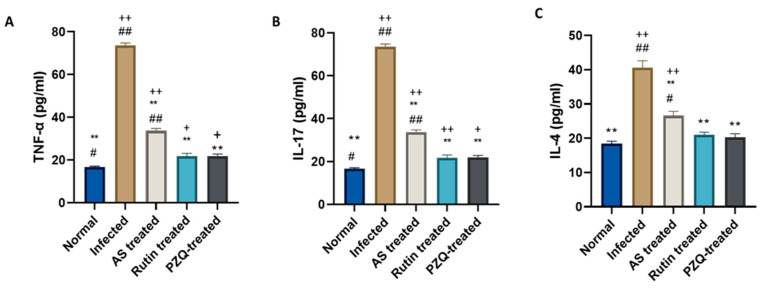
Effect of treatment with rutin naturally isolated from *A. sativum* on serum cytokines. Effect of different treatments on (**A**) TNF-α, (**B**) IL-17, (**C**) IL-4. Data expressed as mean ± SE (*n* = 10). *P*-value represented the relationship between ASb- and rutin-treated groups and normal control, *S. mansoni*-infected controls and drug controls. ** *p*-value < 0.001, compared with infected control group; + *p* < 0.05, ++ *p*-value < 0.001 in comparison to normal control group; # *p*< 0.05, ## *p*-value < 0.001 in comparison to PZQ control group.

**Figure 5 nutrients-15-01206-f005:**
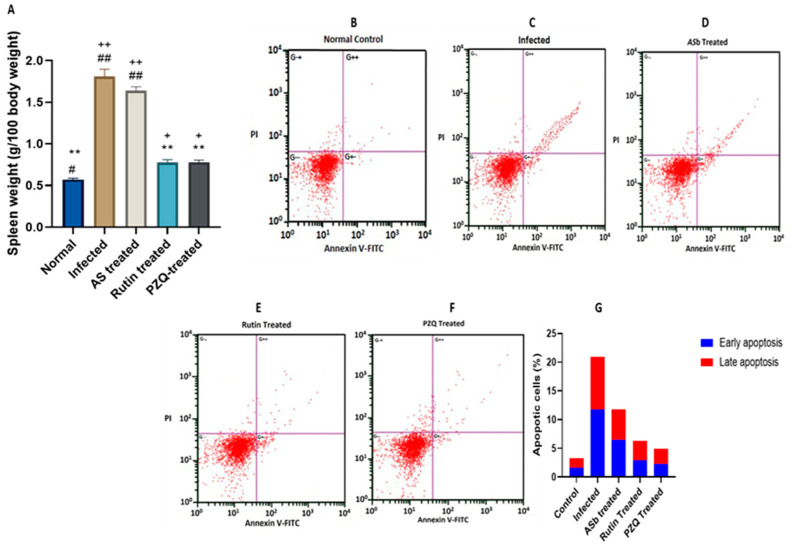
Effect of treatment with rutin naturally isolated from A. sativum on apoptosis in the spleen caused by infection with 100 ± 2 *S. mansoni* cercariae, 8 weeks postinfection. (**A**) Spleen weight (relative to body weight); (**B**–**F**) typical flow cytometry plots showing annexin V-FITC and PI double staining. The figure shows live cells in the bottom left quadrant (double negative); early apoptotic cells are displayed in the bottom right quadrant (Annexin V positive and PI negative), while late apoptotic cells are displayed in the top right quadrant (double-positive). (**G**) Percentages of viable cells, shown as the mean ± SE (*n* = 5). ** *p*-value < 0.001, + *p* < 0.05, ++ *p*-value < 0.001, # *p*< 0.05, ## *p*-value < 0.001.

**Table 1 nutrients-15-01206-t001:** Experimental design.

Group I: Normal Control	Uninfected and Untreated Mice.
Group II: Infected control	Infected with 100 ± 2 *S. mansoni* cercariae/mouse via the tail; act as infected control group.
Group III (AS treated)	Mice were infected by 100 ± 2 *S. mansoni* cercariae/mouse and treated with 50 mg/kg body weight *AS*b ethanolic extract [21] single dose per day for 40 days.
Group IV (Rutin treated)	Mice were infected by 100 ± 2 *S. mansoni* cercariae/mouse and received 40 mg/kg body weight rutin, single dose per day for 40 days.
Group V (PZQ-treated)	Mice were infected by 100 ± 2 *S. mansoni* cercariae/mouse and treated with 500 mg (dissolved in 70% glycerin)/kg of body weight PZQ; orally 7 weeks postinfection for two successive days [31].

**Table 2 nutrients-15-01206-t002:** UV and ^1^H-NMR spectral data of rutin compound.

UV Spectral Data (λ_max_, nm)	
Reagent	(λ_max_, nm)
MeOH	258, 329, 358 (flavonol)
NaOMe	272, 329 (sh), 407 (free of OH at 3′ and 4′)
NaOAc	272, 355 (sh), 393 (free OH at 7)
NaOAc + H_3_BO_3_	260, 379 (ortho-dihydroxy group at B ring)
AlCl_3_	275, 422 (free OH at 5)
AlCl_3_ + HCl	272, 396 (ortho-dihydroxy group at B ring).
1H-NMR Spectral Data δ (ppm)	
7.57 (1H, d, J = 2.1 Hz, H-2′), 7.54 (1H, d, J = 9, 2.1 Hz, H-6′),6.89 (1H, d, J = 9.0 Hz, H-5′), 6.40 (1H, d, J = 2.1 Hz, H-8), 6.20 (1H, d, J = 2.1 Hz, H-6), 5.35 (1H, d, J = 7.5 Hz, anomeric cH-1″, glucose), 4.40 (H, broad singlet, anomeric H-1‴, rhamnose), 3.25–3.45 (m, the rest sugar of glucose and rhamnose), 1 (3H, d, J = 6.3 Hz, CH3- rhamnose).	

**Table 3 nutrients-15-01206-t003:** Effect of different treatments on parasitological parameters of *S. mansoni*-infected CD1 mice.

Mice Groups	Worm Burden		Liver Egg Counts		Feces Egg Counts	
	Mean ± SE	Reduction (%)	Mean ± SE	Reduction (%)	Mean ± SE	Reduction (%)
Infected control	29.18 ± 6.95	0	5170 ± 809	0	4532 ± 598	0
*AS*-treated	11.64 ± 3.69 ^b^	60.11	2720± 122 ^b^	47.39	1814 ± 273 ^b^	59.97
Rutin-treated	9.3 ± 1.84 ^c^	68.13	758 ± 33 ^c^	85.34	942 ± 531 ^c^	79.21
PZQ-treated	1.3 ± 0.92 ^c^	95.54	129 ± 10 ^c^	97.50	655 ± 131 ^c^	85.55

CD1 mice were infected with 100 ± 2 *S. mansoni* cercariae/mouse via the tail. Worm burden and egg load were estimated at 8 weeks postinfection. Data were represented as mean ± SE (*n* = 5) and are typical of two independent experiments. ^b^
*p* < 0.01 with respect to the infected control group; ^c^
*p* < 0.001 with respect to the infected control group.

**Table 4 nutrients-15-01206-t004:** Histopathological analysis in mice after different treatment.

Parameters ^a^	Normal	Infected	AS-Treated	Rutin-Treated	PZQ-Treated
Inflammatory infiltration	0.00 ± 0.00	2.80 ± 0.20	2.50 ± 0.54	2.10 ± 0.57	2.0 ± 0.5
Fibers accumulation	0.00 ± 0.00	2.40 ± 0.55 ^b^	1.90 ± 0.55	1.20 ± 0.47 ^a^	1.60 ± 0.54 ^a^
Tissue damage	0.00 ± 0.00	2.60 ± 0.53 ^b^	2.10 ± 0.55	1.40 ± 0.54 ^a^	1.20 ± 0.47 ^a^
Necrosis	0.00 ± 0.00	2.30 ± 0.55 ^b^	1.80 ± 0.55	1.20 ± 0.83 ^a^	1.30 ± 0.45 ^a^

Histopathological lesions were scored in accordance with their severity. Data expressed as mean ± SE (*n* = 5). *p*-value represents the relationship between AS- and rutin-treated groups and *S. mansoni -*infected controls and PZQ controls. ^a^ statistically significant difference from the infected control group, ^b^ statistically significant difference from the drug control group.

## Data Availability

The data presented in this study are available in the article and Appendix A.

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
