# Peer review of "Rutin, a Flavonoid Compound Derived from Garlic, as a Potential Immunomodulatory and Anti-Inflammatory Agent against Murine Schistosomiasis mansoni"

_nutrients, 2023, doi:10.3390/nu15051206_

Round 1
Reviewer 1 Report
This manuscript described the effect of Allium sativum (garlic) bulbs extract, rutin and Praziquantel on schistosomiasis induced by Schistosoma (S.) mansoni. The results showed that the treatment with 40 mg/kg of rutin showed anti-schistosomiasis effect through immunoregulation and anti-inflammation. The whole research idea is clear, but there are still some problems in the design and paper writing.
Thus, I think this manuscript could be published after major revision.
Overall, the article should undergo major revisions in the Results and Discussion sections. In the Results section, there are too many descriptions of the specific values in the figures and tables, and these information that can be directly obtained by readers in the figures should be deleted. In addition, a summary description of the trial results and explanations of relevant results were lacking. The contents of the Discussion section and the Results section repeat too much and are too long. The relevant contents should be combined together and discussed as a whole to make the article more concise.
In addition, the color, style and layout of the pictures in the text are quite different, which should be kept unified. Although the number of pictures meets the requirements, they should be more concise. For example, the structure diagram of rutin can be put into Figure 1, and Figure 2 can be merged with Figure 1. Figure 4 and Figure 5 are liver-related indicators, which can be combined. Both Figures 7 and 6 are related to the spleen, and Figure 7 can be incorporated into Figure 6.
1. In the studies determining the intragastric dose of rutin, the reasons for the selection of the dose of 40 mg/kg/day were not clearly described in the article, so figures or tables of the relevant trial results should be supplemented.
2. In the description of Table 1, the dose of ASb used "50 mg/kg body weight" is from the reference. Is the intake of Rutin in ASb at this dose consistent with the intake in Group IV(Rutin treated)? If not, what is the rationale for choosing this dose in this study?
3. After the separation and identification of ASb ethanolic extract, it is suggested to measure the content of Rutin in the extract in order to better compare the test results between Group II(AS treated) and Group IV(Rutin treated).
4. In the study of inflammatory factors, in addition to measuring the content of TNF-α, IL-17 and IL-4 in serum, it is recommended to measure the gene and protein expression of related factors in liver and spleen tissues.
5. In the description in line 153, the mice were killed under ether anesthesia 8 weeks after injection of Schistosoma (S.) manson, but in the description in Table 1, both Group II(AS treated) and Group IV(Rutin treated) were treated by gavage for 40 days. For what period of time was no gavage treatment performed? Why not do it?
6. Line 78: The size of the sieve should be indicated to clarify the particle size of the ground powder.
7. Line 81: "until the solution became clear" is vague, and the specific extraction time should be clearly written.
8. Additional information on the weeks of age of albino CD-1 mice used in the study.
9. Figure 2 is not clear enough, it is recommended to use a picture with higher definition.
10. There are many font formatting problems in the text. The font of the title, body, figure note and table note should be consistent, such as line 229-230: the body part does not need to be italicized; line 270-273: Note font should be different from the body.
11. The species name of Schistosoma (S.) mansoni should be italicized, but it is not italicized in many places in this article.
12. The format of p-values in the text is not uniform and should be in lowercase italics.

Reviewer 2 Report
Dear Author,
Congratulations on a very interesting paper. My comments on the work are only related to the methodology of the study - the determination of the flavonoid content and the way the literature items are presented.
In the materials and methods section when describing the method of determination of total flavonoid content [line 84] you refer to a literature item on Alfwuaires et al., [26], which does not give information on how the study was performed...as it refers to another item - i.e. it is not a source item. I would very much appreciate if you could post the correct literature item or describe the methodology.
In literature items 12,23,39,44 all authors are not listed. I know that sometimes the number of authors is large, but unfortunately they should all be listed in the citation.
These minor remarks do not affect the value of the paper, but definitely tidy it up.
